# Comparison of GF2 and SPOT6 Imagery on Canopy Cover Estimating in Northern Subtropics Forest in China

**Jingjing Zhou [1,2], Yuanyong Dian [1,2,*], Xiong Wang [1], Chonghuai Yao [1], Yongfeng Jian [1], Yuan Li [1] and Zeming Han [1]**

[1] College of Horticulture and Forestry, Huazhong Agricultural University, Wuhan 430070, China; hupodingxiangyu@mail.hzau.edu.cn (J.Z.); hzauwangxiong@163.com (X.W.); yao_chonghuai@mail.hzau.edu.cn (C.Y.); jianyongfeng@163.com (Y.J.); 18404984003@163.com (Y.L.); swxfhzm@163.com (Z.H.)

[2] Hubei Engineering Technology Research Centre for forestry Information, Huazhong Agricultural University, Wuhan 430070, China

\* Correspondence: dianyuanyong@mail.hzau.edu.cn; Tel.: +86-027-8728-2010

**Abstract:** Canopy cover is an important vegetation attribute used for many environmental applications such as defining management objectives, thinning and ecological modeling. However, the estimation of canopy cover from high spatial resolution imagery is still a difficult task due to limited spectral information and the heterogeneous pixel values of the same canopy. In this paper, we compared the capacity of two high spatial resolution sensors (SPOT6 and GF2) using three ensemble learning models (Adaptive Boosting (AdaBoost), Gradient Boosting (GDBoost), and random forest (RF)), to estimate canopy cover (CC) in a Chinese northern subtropics forest. Canopy cover across 97 plots was measured across 41 needle forest plots, 24 broadleaf forest plots, and 32 mixed forest plots. Results showed that (1) the textural features performed more importantly than spectral variables according to the number of variables in the top ten predictors in estimating canopy cover (CC) in both SPOT6 and GF2. Moreover, the vegetation indices in spectral variables had a lower relative importance value than the band reflectance variables. (2) GF2 imagery outperformed SPOT6 imagery in estimating CC when using the ensemble learning model in our data. On average across the models, the $R^2$ was almost 0.08 higher for GF2 over SPOT6. Likewise, the average RMSE and average MAE were 0.002 and 0.01 lower in GF2 than in SPOT6. (3) The ensemble learning model showed good results in estimating CC, yet the different models performed a little differently in the results. Additionally, the GDBoost model performed the best of all the ensemble learning models with $R^2 = 0.92$, root mean square error (RMSE) = 0.001 and mean absolute error (MAE) = 0.022.

**Keywords:** GF2; SPOT6; high spatial resolution; canopy cover; ensemble learning model; gray level co-occurrence matrix (GLCM)

## 1. Introduction

Forest canopy cover (CC) is a widely used indicator to characterize the structure and functioning of forest ecosystems. Additionally, canopy cover is an important statistical variable for forest resource management, planning, and reporting on national and international scales. According to the Food and Agriculture Organization (FAO), CC is defined as the proportion of ground covered by the vertical projection of the tree crowns [1], including small gaps inside the crown perimeter [2]. Accordingly, approaches to quickly and efficiently estimate CC, especially over broad areas, is important. Recently, with the development of remote sensing sensors, researchers are interested in mapping regional

or global forest canopy cover with various remotely sensed data to provide inputs for land cover monitoring or ecosystem modeling [3–6].

To date, medium spatial and spectral resolution imageries have been widely used in CC estimating, such as Landsat series imageries and moderate resolution imaging spectroradiometer (MODIS) imageries [4,7–11]. However, mixed pixels are common in heterogeneous land cover, which can result in inaccurate estimates of CC [4,12–15]. The launch and availability of sensors with improved spatial, spectral, and radiometric characteristics can further contribute to improving the accuracy of forest measuring variables. SPOT-6/7 series sensors with a high spatial resolution (1.5 m in panchromatic and 6.0 m in multispectral bands) have been used in forest variable prediction and achieved encouraging results in estimating forest variables [16]. The Gaofen-2(GF-2) satellite, which launched in 2014 and is configured with two panchromatic and multispectral charge coupled device (CCD) camera sensors, can achieve a spatial resolution of 1 m in panchromatic mode and a resolution of 4 m in four spectral bands in multispectral mode [17–19]. Moreover, it is also characterized by high radiative accuracy, high positioning accuracy, and fast attitude maneuverability, among other features.

Meanwhile, a large number of variables are derived from multispectral imagery to estimate CC. The spectral variables are the most used variables, such as the normalized difference index (NDVI), the normalized difference index using the green band (NDVIg), the chlorophyll index using the green band (CIg), the enhanced vegetation index (EVI) and the soil adjusted vegetation index (SAVI). These spectral variables were widely used in estimating forest variables such as canopy cover, biomass, and leaf area index (LAI) [20–24]. Moreover, the contextual variables which indicated the pattern of spatial distributions of gray were performed to promote the accuracy in estimating forest parameters [25–27]. The gray level co-occurrence matrix was the conventional textural variable which was widely used in image analysis.

The most common method to estimate CC from remote sensing imagery is the regression-based method [7,9,11,13,20,28–32]. Recently, with the development of machine learning algorithms, some ensemble learning methods have been gradually used in regression, such as the boosting and bagging strategies which can obtain a boost in accuracy [7,27]. Boosting is a method that builds multiple models, each model learns to fix the prediction errors of a prior model in the sequence of models. The two most common boosting ensemble algorithms are Adaptive Boosting and Gradient Boosting. Bagging is another method that builds multiple models from different subsamples of the training dataset. Random forest is a common method of bagging algorithm. These ensemble learning methods help us to improve the accuracy in estimating forest variables with remote sensing data.

Our main objective of this paper is to compare the new GF2 multispectral images with SPOT6 images in estimating CC. Moreover, we aim to further compare the three ensemble learning methods by using bagging and boosting strategies in estimating canopy cover in the northern subtropics zone forest with different forest types.

## 2. Materials and Methods

### 2.1. Study Area

The study was conducted at Taizi Mountain (112°48′E–113°03′ E, 30°48′N–31°02′ N) in Hubei Province, China, which is also designated as a national forestry park (as seen in Figure 1). The region covers approximately 7600 ha, and the percentage of forest cover is 85%. The study site is in the north subtropical humid monsoon climate zone with a mean annual rainfall of 1094.66 mm and a mean annual temperate of 16.4 °C. The majority of topographic features in this area are low mountains and hills. Elevation ranges from 40.3 m along the valley bottom to 467.4 m on the low mountain tops. Soils are generally characterized as yellow-brown earth. Plantation Chinese red pine (*Pinus massioniana* Lamb.) trees are the dominate species in this area, while, there are also some other broadleaf forest species such as sawtooth oak (*Quercus acutissima* Carr) and trident maple (*Acer buergerignum* Mig.) Three different woodland types are described in the research area: needle forest (the main species is

*Pinus massioniana*), broadleaf forest and mixed forest (broadleaf and needle mixed). The area ratios of needle forest, broadleaf forest, and mixed forest to all forested areas were 42.3%, 24.7%, and 33.0%, respectively according to statistical data from the local forest management department.

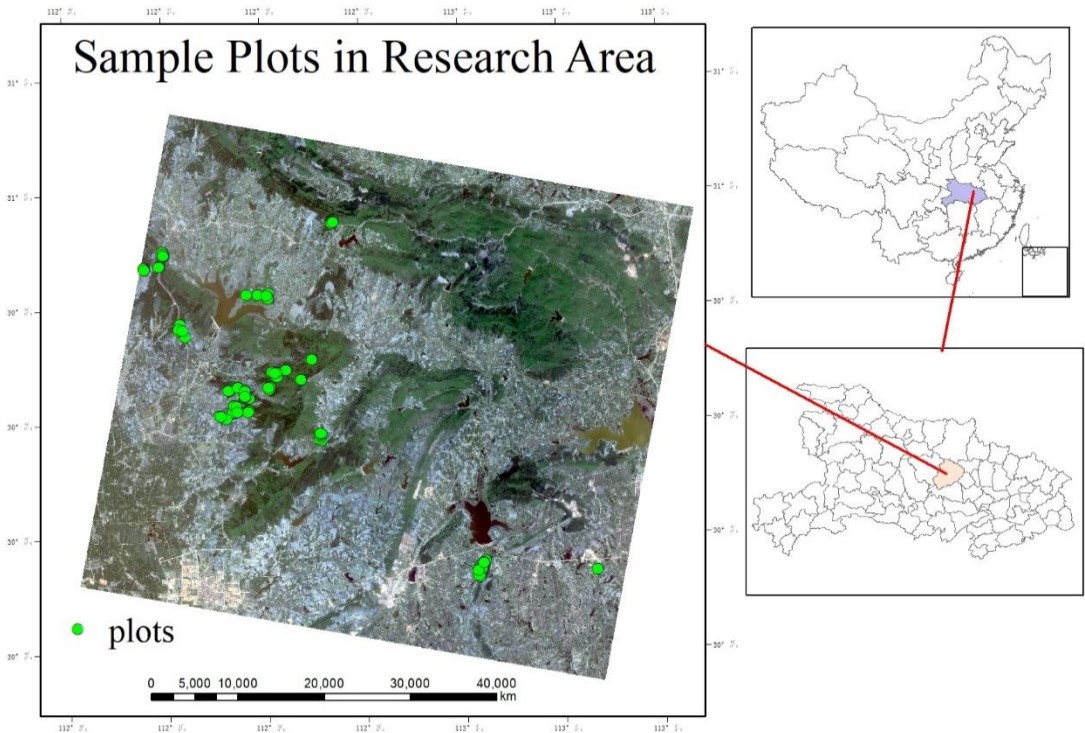

**Figure 1.** Location of the research area and the filed plots identified in GF2 imagery.

## 2.2. Data

### 2.2.1. Field Data

In total, we derived 97 plots in this area based on the stratified sampling method according to the forest types in August 2015 and there were 41 plots of needle forest, 24 plots of broadleaf forest and 32 plots of mixed forest. The number of plots with different forest types was determined according to the area ratios of forest type area to total forested area. The plot size was 20 m × 20 m. At each plot, we recorded species, DBH (diameter at breast height), tree height, x, y location (m) of the tree base relative to the plot starting point, canopy width in North-South (NS) and West-East (WE) direction. At the plot level, woodland types and the GPS location of starting points were recorded. The position of plots was located by a differential GPS unit (Trimble GeoXH6000 GPS units) and corrected with high precision real-time differential signals received from Hubei Continuously Operation Reference Stations (HBCORS). The location error was less than 1 m, which allowed for the plots to be effectively geo-referenced with satellite data. Canopy shape was assumed as an ellipse and canopy widths in NS and WE direction were regarded as major semi-axis and minor semi-axis, respectively. The canopy cover area was calculated by accumulating each tree crown project area in the plot and then clipped by the plot boundary. Percent canopy cover was defined as a crown projection area in the plot divided by the plot area (as seen in Figure 2). The one-way analysis of variance (ANOVA) and a further statistical method of least significant difference (LSD) utilizing multiple comparison tests to estimate the significant effects of forest type on the CC.

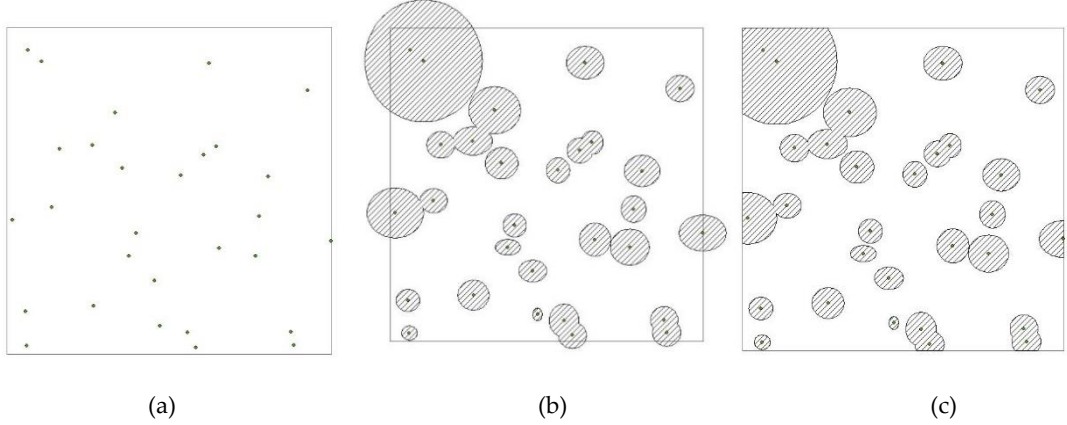

|  (a)  |  (b)  |  (c)  |

**Figure 2.** Canopy cover per plot, which was determined by the tree location and crown diameter in NS and WE direction (**a**) tree location (**b**) tree canopy cover area (**c**) canopy cover in a plot.

### 2.2.2. Remote Sensing imagery

SPOT6 and GF2 imageries were acquired in August 2015 almost consistent with the field investigation. We converted the SPOT6 and GF2 imagery into surface reflectance using the FLAASH algorithm in ENVI 5.4(Exelis Visual Information Solutions, Boulder, CO, USA). A geometrical correction was applied before atmospheric correction. The root mean square error of geometrical correction was 1.83 m, which was lower than 1 pixel. As the SPOT6 image does not have the same spatial resolution with GF2, we resampled the SPOT6 image pixel size to 4 m × 4 m which was the same as the GF2. Resampling was done with a 'nearest neighbor' technique. The spectral and spatial parameters of GF-2 images and SPOT-6 are listed in Table 1.

**Table 1.** The spectral bands and resolution of GF2 and SPOT6.

|  | SPOT6 | | GF2 | |
| --- | --- | --- | --- | --- |
| **Band** | **Wavelength (nm)** | **Resolution (m)** | **Wavelength (nm)** | **Resolution (m)** |
| Blue (B) | 455–525 | 6.8 | 450–520 | 4 |
| Green (G) | 530–590 | 6.8 | 520–590 | 4 |
| Red (R) | 625–695 | 6.8 | 630–690 | 4 |
| Near infrared (NIR) | 760–890 | 6.8 | 770–890 | 4 |
| Panchromatic | 455–745 | 1.5 | 450–900 | 1 |

### 2.2.3. Predictor Variables

In this paper, a total of 17 predictor variables were selected in predicting forest parameters, including 9 spectral variables and 8 textural variables. Spectral variables were extracted from the GF2 and SPOT6 multispectral images including 4 reflectance bands (b1–blue, b2–green, b3–red, and b4–nir) and 5 common spectral vegetation indices (VIs). These VIs were the normalized difference index (*NDVI*), the normalized difference index using the green band (*NDVIg*), the Chlorophyll index using the green band (*CIg*), the enhanced vegetation index (*EVI*), and the soil adjusted vegetation index (*SAVI*). According to the spatial resolution and plot size, we derived each variable with the median value of a 5 × 5 window. The window location was determined by the plot position in the satellite imagery. The VIs and its calculation formula are listed in Table 2.

**Table 2.** Vegetation indices and its formula.

| Vegetation Indices (VIs) | Formula |
|---|---|
| 1. Normalized difference vegetation index (*NDVI*) | $NDVI = (nir - r)/(nir + r)$ |
| 2. Normalized difference index using the green band (*NDVIg*) | $NDVI_g = (nir - g)/(nir + g)$ |
| 3. Chlorophyll index using the green band (*CIg*) | $CI_g = nir/g - 1$ |
| 4. Enhanced Vegetation Index (*EVI*) | $EVI = 2.5 \times (nir - r)/(nir + 6 \times r - 7.5 \times b + 1)$ |
| 5. Soil Adjusted Vegetation Index (*SAVI*) | $SAVI = (1 + L)(nir - r)/(nir + r + L)$ |

Notes: $b$, $r$, and *nir* represent reflectance in the blue, red, and near-infrared wavelengths, respectively. Parameter $L$ represents the SAVI term (set to 0.5).

The gray level co-occurrence matrix (GLCM) was a powerful approach and widely used for image texture analysis. In this paper, we derived eight texture parameters—the mean (MEAN), homogeneity (HOM), contrast (CON), dissimilarity (DIS), Entropy (ENT), variance (VAR), angular Second Moment (ASM), and correlation (COR)—from the GLCM (see Table 3) based on the first principal component of multispectral bands in SPOT6 and GF2 imagery. Texture parameters are sensitive to the window size of GLCM. Generally, the window size was determined by the spatial resolution and the ground object size. According to the research results of Zhao et al. (2018) [27], the window size was set to $9 \times 9$ and the gray level was set to 256 in this paper. The distance and direction between the reference pixel and the neighbor pixel in GLCM were set to 1 and diagonally down, respectively.

**Table 3.** Formula of textural variables used in this study.

| Gray Level Co-Occurrence Matrix based Texture Parameter Estimation | Formula |
|---|---|
| 1. Mean (MEAN) | $MEAN = \frac{1}{N^2} \sum\limits_{i,j=0}^{N-1} P_{i,j}$ |
| 2. Homogeneity (HOM) | $HOM = \sum\limits_{i,j=0}^{N-1} i \frac{P_{i,j}}{1+(i-j)^2}$ |
| 3. Contrast (CON) | $CON = \sum\limits_{i,j=0}^{N-1} iP_{i,j}(1-j)^2$ |
| 4. Dissimilarity (DIS) | $DIS = \sum\limits_{i,j=0}^{N-1} iP_{i,j}\left|1-j\right|$ |
| 5. Entropy (ENT) | $ENT = \sum\limits_{i,j=0}^{N-1} iP_{i,j}\left(-\ln P_{i,j}\right)$ |
| 6. Variance (VAR) | $VAR = \frac{\sum_{i,j}(X_{ij}-\mu)^2}{n-1}$ |
| 7. Angular Second Moment (ASM) | $ASM = \sum\limits_{i,j=0}^{N-1} iP_{i,j}{}^2$ |
| 8. Correlation (COR) | $COR = \frac{\sum_{i,j=0}^{N-1} iP_{i,j}-\mu_1\mu_2}{\sigma_1^2\sigma_2^2}$ $\mu_1 = \sum\limits_{i=0}^{N-1} i \sum\limits_{j=0}^{N-1} P_{i,j}$ $\mu_2 = \sum\limits_{j=0}^{N-1} j \sum\limits_{j=0}^{N-1} P_{i,j}$ $\sigma_1^2 = \sum\limits_{i=0}^{N-1} (i-\mu_1)^2 \sum\limits_{j=0}^{N-1} P_{i,j}$ $\sigma_2^2 = \sum\limits_{j=0}^{N-1} (j-\mu_2)^2 \sum\limits_{j=0}^{N-1} P_{i,j}$ |

*2.3. Modeling Methods*

In this paper, we selected the ensemble learning models to predict canopy cover, which included the boosting method and the bagging method.

Machine learning methods were widely used to solve the classification and regression problems. There are so many algorithms in regression, such as the classification and regression tree (CART) and support vector machine (SVM). However, each algorithm has its owns advantages and disadvantages. The ensemble method helps improve machine learning results by combining several models, which is going to be used widely in many situations.

Ensemble learning methods are meta-algorithms that combine several machine learning techniques into one predictive model in order to decrease variance and bias [33]. Ensembles will help us to promote accuracy. There are two popular methods used in ensemble methods: bagging and boosting. The bagging method is building multiple models (typically of the same type) from different subsamples of the training dataset. The random forest (RF) method is a state-of-the-art bagging method, which builds several decision trees independently and then averages their predictions. By averaging, the combined result was usually better than any single decision tree. While, the boosting method builds multiple base estimators (the decision tree was used in this paper), each of which learns to fix the prediction errors of a prior model in the sequence of models, like the adaptive boosting (AdaBoost) method and gradient boosting (GDBoost). The difference between AdaBoost and GDBoost is how they work with the underfitted values of their predecessors. The AdaBoost algorithm modifies the weight of the underfitted training instances by the previous training at every interaction, while the GDBoost algorithm tries to fit the new predictor to the residual errors made by the previous predictor.

In this paper, we choose the RF, AdaBoost, and GDBoost methods to compare the effect of these two ensemble strategies in CC prediction. All the algorithms were implemented in the scikit-learn 0.22.2 package in python 3.7 [34]. To optimize the parameters in ensemble models, we used the grid search approach to tune the parameters, which evaluated a model accuracy for each combination of algorithm parameters specified in a grid. According to the instructions of ensemble algorithms in scikit-learn, we tuned different parameters in these models. The number of trees (*ntree*) and learning rate (*learning_rate*) were tuned in AdaBoost and GDBoost models, while the number of trees (*ntree*) and size of random subsets of features to consider in constructing trees (*max_features*) were tuned in the RF model.

## 2.4. Model Evaluations

In order to assess the performance of all models in canopy cover predicting, we split the field data into two parts—70% for training and 30% for testing. The coefficient of determination ($R^2$), root mean squared error (*RMSE*), and mean absolute error (*MAE*) were used to assess the best fit of the predicted and field measured canopy cover. The $R^2$, RMSE, and MAE can be calculated from the following equations:

$$R^2 = 1 - \sum_{i=1}^{n} (y_i - \hat{y}_i)^2 / \sum_{i=1}^{n} (y_i - y_m)^2 \tag{1}$$

$$RMSE = \sqrt{\sum_{i=1}^{n} (y_i - \hat{y}_i)^2 / n} \tag{2}$$

$$MAE = \frac{1}{n} \sum_{i=1}^{n} | y_i - \hat{y}_i| \tag{3}$$

## 3. Results

### 3.1. Canopy Cover Characteristics in Different Forest Types

The field data characteristics are summarized in Table 4. The CC value showed that there was a great variation in all forest types in our dataset. The coefficient variation (CV) in the mixed forest was higher than in the needle and broadleaf forests. The results of ANOVA found that there was a significant difference in CC, and the further statistical method of the LSD multiple comparison tests

showed that each forest type's CC was significantly different. These results indicated that there was a significant influence of forest type on CC.

**Table 4.** Canopy cover characteristics of each forest type.

| Species Type | Number | Mean of Canopy Cover (CC) | Std of CC | Coefficient Variation |
|---|---|---|---|---|
| Needle forest[a] | 41 | 0.623 | 0.128 | 20.55% |
| Broadleaf forest[b] | 24 | 0.770 | 0.110 | 14.29% |
| Mixed forest[c] | 32 | 0.564 | 0.120 | 21.28% |

Note: different letters indicate significant differences($p < 0.05$) among forest types based on one-way ANOVA followed by an LSD test (F = 20.939, $p < 0.000$).

### 3.2. Assess the Relative Importance Value of Predictors for the Three Models

We first assessed the relative importance value (RIV) of the 17 predictor variables used in the AdaBoost, GDBoost and the RF algorithm to find which variables were sensitive to CC estimating. Because the decision tree was the base estimator of these three ensemble methods, the relative rank of a feature used as a decision node in a tree can be used to assess the relative importance of that feature connecting the predictability of the target variable. In the scikit-learn package [34], the relative importance value of the three ensemble models can be calculated with the ensemble package, the results are listed in Figure 3. For SPOT6 imagery, we found that the g band was the first important variable whether in the AdaBoost, GDBoost, or RF models, and it was also higher than any other variable. Besides, the textural variables were the second most important features in all three models. But there was a little difference in the boosting and bagging approaches, the COR feature was second in the AdaBost and GDBoost models, and the VAR feature was second in the RF model. The EVI, NDVIg, and NDVI had the lowest RIV in the AdaBoost, GDBoost, and the RF model, respectively.

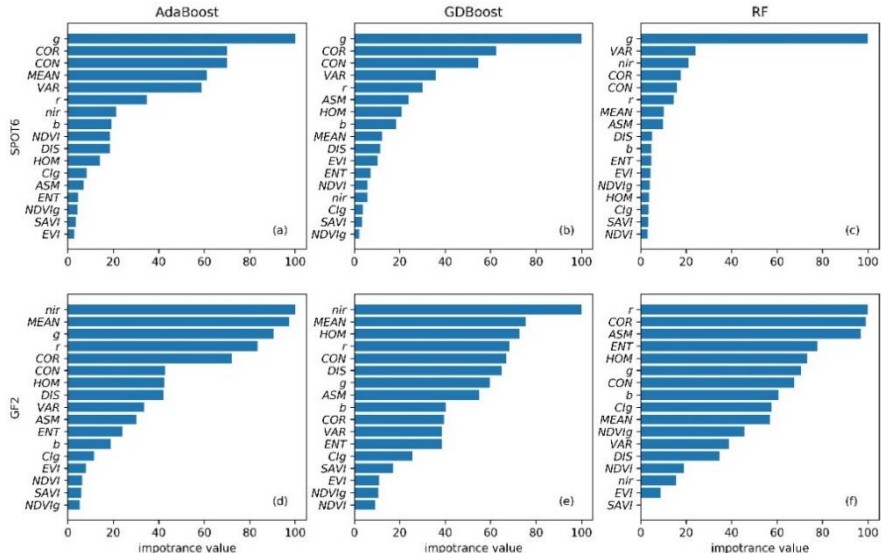

**Figure 3.** Relative importance value (RIV) of different AdaBoost, GDBoost and RF models in SPOT and GF2 (**a**) RIV based on AdaBoost in SPOT6 (**b**) RIV based on GDBoost in SPOT6 (**c**) RIV based on RF in SPOT6 (**d**) RIV based on AdaBoost in GF2 (**e**) RIV based on GDBoost in GF2 (**f**) RIV based on RF in GF2. (*b*, g, *r*, and *nir* represent reflectance in the blue, green, red, and near-infrared wavelengths, respectively, *NDVI, NDVIg, CIg, EVI,* and *SAVI* represent vegetation indices, *MEAN, HOM, CON, DIS, ENT, VAR, ASM*, and *COR* represent texture parameters derived from gray level co-occurrence matrix (GLCM)

In contrast to SPOT6, GF2 was different in the relative importance value of the input variables. The nir band had the highest RIV in AdaBoost and GDBoost, yet the r band had the highest RIV in the RF model. The contextual variable MEAN was the second important variable in AdaBoost and

GDBoost, while the COR was the second important variable in the RF model. The NDVIg, NDVI, and SAVI had the lowest RIV in AdaBoost, GDBoost, and the RF model, respectively.

In general, the number of textural features were more important than the spectral variables in the top ten predictors in both SPOT6 and GF2. Especially, the vegetation indices in spectral variables had a lower relative importance value than the band reflectance variables.

### 3.3. Parameter Tuning for Models

The grid search was done separately for each image and model. In AdaBoost and GDBoost, we tuned two parameters: *ntree* and *learning_rate*, while we tuned *ntree* and *max_features* in the RF model. The *ntree* was tested from 50 to 500 stepped by 50, the *learning_rate* was tested 0.5 to 1.0 stepped by 0.1, and the *max_features* was tested from 2 to 17 stepped by 1. The heat map of $R^2$ corresponding with the search space of different models and images are shown in Figure 4. According to this visual map, we found that in AdaBoost and GDBoost models, the highest $R^2$ was acquired when the *ntree* parameter was set to 150 in all three models and the *learning_rate* was set to 0.9 whether in SPOT6 or GF2 images. While in the RF model, when the *ntree* was set to 150 and *max_features* was set to 10 it would derive the highest $R^2$.

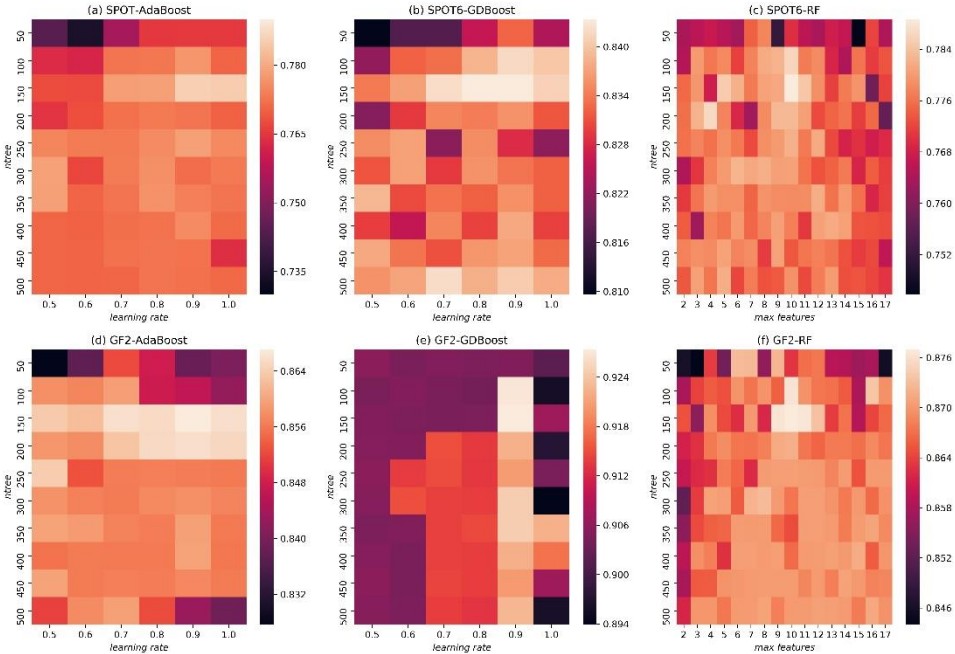

**Figure 4.** The coefficient of determination ($R^2$) heat map of the search space in the grid search with SPOT6, GF2 images, and different models. (**a**) $R^2$ heat map with the AdaBoost model in SPOT6 (**b**) $R^2$ heat map with the GDBoost model in SPOT6 (**c**) $R^2$ heat map with the RF model in SPOT6 (**d**) $R^2$ heat map with the AdaBoost model in GF2 (**e**) $R^2$ heat map with the GDBoost model in GF2 (**f**) $R^2$ heat map with the RF model in GF2. (*ntree* represents number of trees, *learning_rate* represents learning rate and *max features* represents the size of random subsets of features to consider in constructing trees.

### 3.4. Canopy Cover Estimation Accuracy

After tuning in these models, the *ntree* was set to 150 and the *learning_rate* was set to 0.9 in AdaBoost and GDBoost, while the *ntree* was set to 150 and *max_features* was set to 10 in the RF model. The CC results were estimated with the optimized parameters, the results are addressed in Table 5 and Figure 5. We found that all three ensemble learning models showed a good result in estimating CC, yet different models performed a little differently in the results. In contrast to the AdaBoost and RF models, the GDBoost model performed the best in both SPOT6 and GF2 imagery, which had the

highest R$^2$ (0.84, 0.92 in SPOT6 and GF2, respectively), the lowest RMSE (0.003, 0.001 in SPOT6 and GF2, respectively) and MAE (0.019, 0.022 in SPOT6 and GF2, respectively).

**Table 5.** Evaluation performance of different methods in predicting canopy cover.

| Sensor | Method | Coefficient of Determination (R$^2$) | Root Mean Square Error (RMSE) | Mean Absolute Error (MAE) |
|---|---|---|---|---|
| SPOT6 | AdaBoost | 0.78 | 0.004 | 0.046 |
| | GDBoost | 0.84 | 0.003 | 0.019 |
| | RF | 0.79 | 0.003 | 0.042 |
| GF2 | AdaBoost | 0.86 | 0.002 | 0.038 |
| | GDBoost | 0.92 | 0.001 | 0.022 |
| | RF | 0.88 | 0.002 | 0.030 |

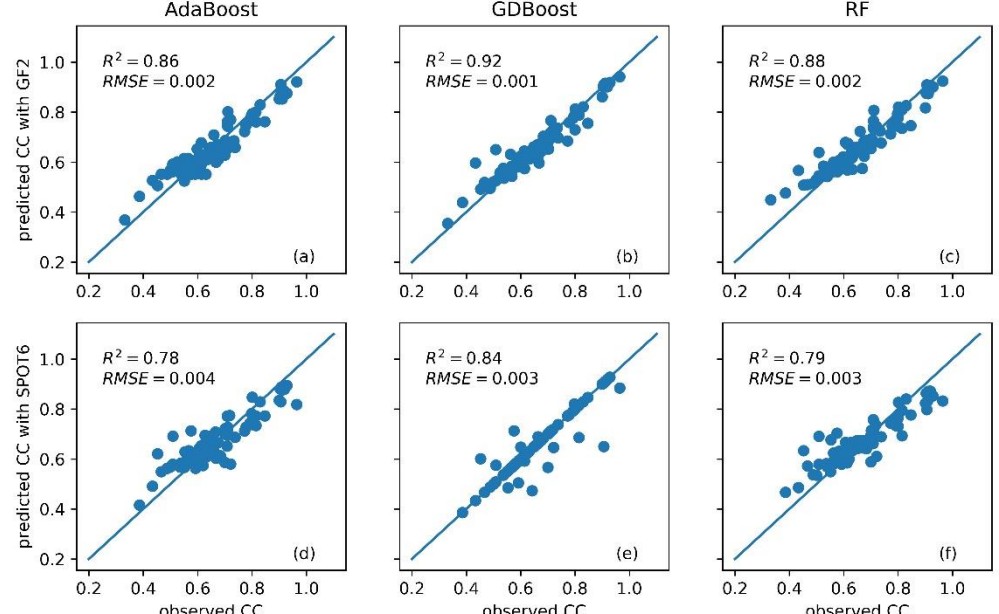

**Figure 5.** Observed vs. predicted canopy cover using SPOT6 and GF2 imagery with different regression models. The solid line indicates the 1:1 correlation between observed and predicted values. (**a**) results of the AdaBoost model in GF2 (**b**) results of the GDBoost model in GF2 (**c**) results of the RF model in GF2 (**d**) results of the AdaBoost model in SPOT6 (**e**) results of the GDBoost model in SPOT6 (**f**) results of the RF model in SPOT6. (CC represents canopy cover, AdaBoost represets Adaptive boosting method, GDBoost represents Gradient boosting model, and RF represents random forest. *R*$^2$ represents coefficient of determination, *RMSE* means root mean square error.)

In comparing with the AdaBoost and RF models, we found that the accuracy of the RF and AdaBoost models was quite similar in both SPOT6 and GF2 imagery. In SPOT6 imagery, the R$^2$ was 0.78 and 0.79 in AdaBoost and RF, respectively. In GF2 imagery, the R$^2$ was 0.86 and 0.88 in AdaBoost and RF, respectively.

In contrast to SPOT6, GF2 performed better in estimating CC in this study than SPOT6. Whether using the AdaBoost, GDBoost or RF models, the accuracy in GF2 outperformed SPOT6. On average across the models, the R$^2$ was almost 0.08 higher for GF2 over SPOT6. Likewise, the average RMSE and average MAE were 0.002 and 0.01 lower in GF2 than in SPOT6.

### 3.5. Mapping Results

The GDBoost model was used to calculate pixel-based CC values from the GF2 imagery with the corresponding predictor variables. Figure 6 shows the CC distribution map of the research area. In the research area, the average value of CC was 0.63. Most forest CC values ranged from 0.4 to 0.8, which accounts for more than 99% of the study area (Table 6), while the CC value of 51.58% area ranged from 0.4 to 0.6 and 48.32% area ranged from 0.6 to 0.8. The center of the research district presented a good canopy cover (>0.6), while the north-east area showed a relatively lower CC (0.4 to 0.6).

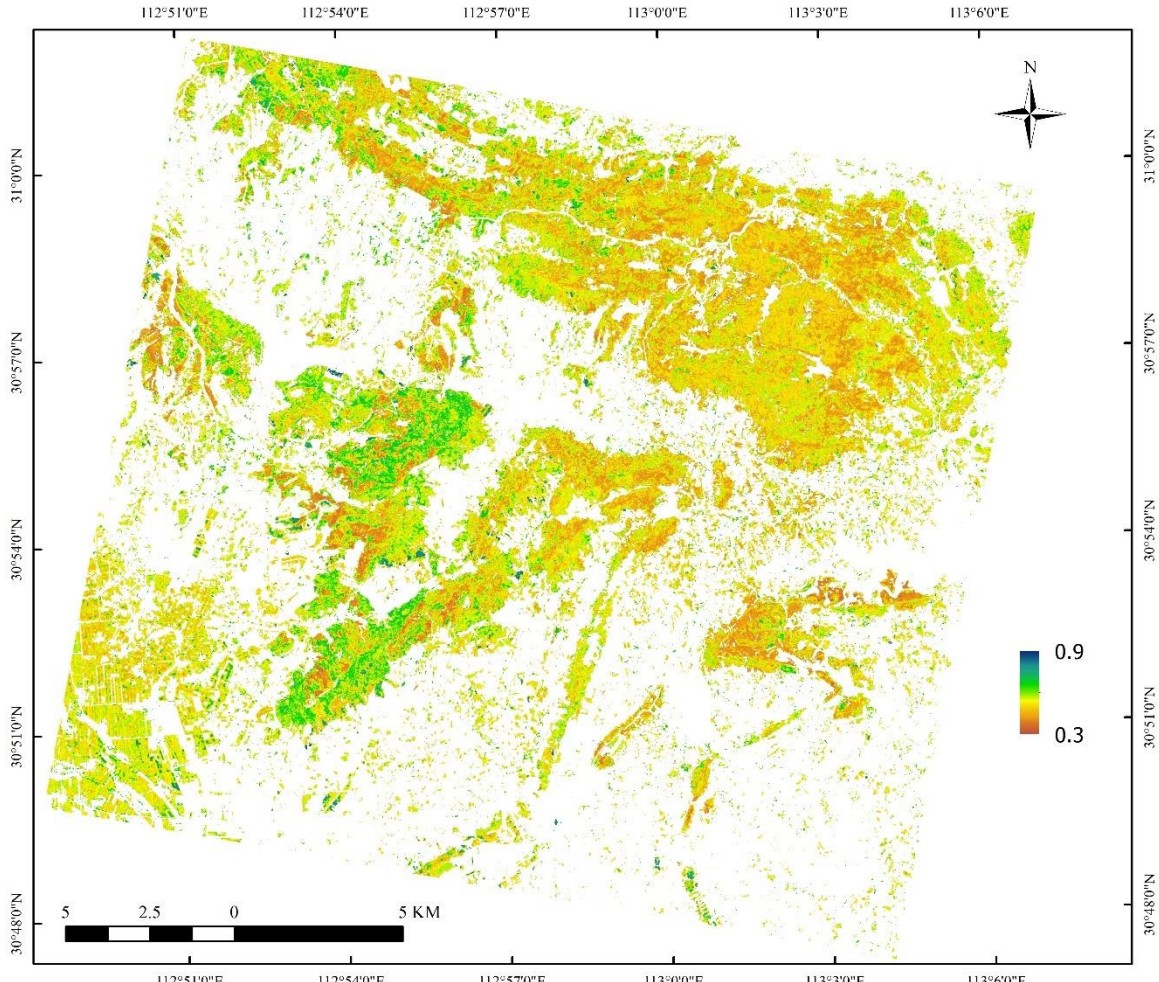

**Figure 6.** Predicted CC map based on the GDBoost model using GF2 imagery.

**Table 6.** Summary of the proportion of different CC levels in the research area.

| CC | Percent (%) |
|---|---|
| <0.4 | 0.03 |
| 0.4–0.6 | 51.58 |
| 0.6–0.8 | 48.32 |
| 0.8–1.0 | 0.07 |

## 4. Discussion

In this study, we compare high spatial resolution imagery GF2 and SPOT6 in the task of estimating canopy cover. Additionally, we test three ensemble models to improve accuracy. Besides, the importance of spectral variables and textural variables in estimating CC are also compared in ensemble models.

High spatial resolution remote sensing imagery with 4 bands (3 bands in the visible and near-infrared bands) performed well at estimating canopy cover variables. Compared to earlier studies on CC estimation in the forest using other types of optical satellite data, the best-case accuracy of GF-2 based CC estimation was equal to or slightly better [6,7,9,30,35]. In this paper, we used the spectral variable and texture content variables derived from GLCM with ensemble models to predict CC and it gave us an encouraging result. We have found that GF-2 imagery provided an encouraging result in estimating canopy cover based on the GDBoost method more than SPOT6 imagery. These results indicated that other high-resolution imageries such as WorldView or Geoeye-1 may have more accuracy in estimating forest variables, but the costs were higher. GF-2 provides a comprehensive and large database of VHR satellite images at an affordable cost, which can be exploited for large scale and different forest types.

We compared the two strategies of ensemble methods—bagging and boosting. The best results were obtained using the GDBoost model. Additionally, the AdaBoost model performed with similar accuracy to the RF model. The results from this study were better than the results presented in Zhao's (2018) [27] and Halperin's (2016) [7] work by using the RF model. The most important reason may be that we used the grid search method to optimize the parameters in the ensemble models. Moreover, the spatial resolution and the tree species may influence the results. Halperin [7] used the Sentinel 2A and Landsat 8 imagery, which had a lower spatial resolution than our data. Zhao [27] used the QuickBird imagery, which had a higher spatial resolution than ours, but it was only focused on Black Locust. In our research, we cannot distinguish tree species.

From the results, it was indicated that the spectral reflectance variables such as r band or nir band seemed to play a more important role than conventional indices (NDVI, NDVIg, CIg, EVI, and SAVI) in estimating CC values. Although these Vis played an important role in the study of vegetation, such as to estimate forest canopy cover [36] and above-ground biomass [37], land cover classification [15], and forest change detection [5,14], it showed a lower relative importance value in the models. Moreover, the texture content was also very important to describe the structure of the forest [7,9], we added these texture variables in the predictor variables which helped us to promote the estimating accuracy due to its high relative importance value in the models. In the heterogeneity forest cover area, the complexity of spatial and vertical structures often obstructed the accurate canopy cover. Adding the texture variable information was a very common method, and our results also found this. The result is in line with Halperin [7], which studied the Nyimba District.

Although many researchers had reported that the short-wave infrared (SWIR) band or red edge band in Landsat or Sentinel 2 sensors help us to acquire a good result in estimating forest variables [6,9], we focused on the high spatial resolution imageries with 4 bands in estimating forest canopy cover. This was because the four bands remote sensing sensors were very common and widely used worldwide, for example, some satellite sensors such as Worldview, Geo-eye, and some airborne sensors such as airborne digital sensor (ADS) 40, digital mapping camera (DMC) 3, and even some unmanned aerial vehicle (UAV) sensors were easy to acquire a large extent of area images. The results in this paper may encourage us to promote utilizing the four bands of imagery.

## 5. Conclusions

Canopy cover is an important forest variable that is used for many purposes. It is a difficult task to estimate canopy cover with high spatial resolution images which contain less spectral information. In this paper, we compared the two high spatial resolution sensors SPOT6 and GF2 across three ensemble learning models in estimating canopy cover. Some insights can be gained through this process. First, the textural features were more important than spectral variables in the top ten predictors in estimating CC in both SPOT6 and GF2. Moreover, the vegetation indices in spectral variables had a lower relative importance value than the band reflectance variables. Second, GF2 outperformed SPOT6 in estimating canopy cover when considering textural features and spectral variables with the ensemble learning model. On average across the models, the $R^2$ was almost 0.08 higher for GF2 over SPOT6. Likewise,

the average RMSE and average MAE were 0.002 and 0.01 lower in GF2 than in SPOT6. Third, the three ensemble learning models provideded good results in estimating CC, yet the different models performed a little differently in the results. The GDBoost model performed the best in all ensemble learning models, and the AdaBoost and RF models performed with similar accuracy in estimating CC in our research.

**Author Contributions:** Conceptualization, Y.D. and C.Y.; methodology, J.Z. and Z.H.; validation, Y.D.,J.Z. and C.Y.; investigation, X.W., Y.J., Y.L. and Z.H.; writing—original draft preparation, Y.D.; writing—review and editing, Y.D. and J.Z.; funding acquisition, Y.D. and C.Y. All authors have read and agreed to the published version of the manuscript.

**Funding:** This research was funded by the National Key Research and Development Program (Grant No. 2017YFC050550404) and National Natural Science Foundation of China (Grant No.51778263).

**Acknowledgments:** We thank Nicholas Coops from the University of British Columbia for assistance on paper structure and helpful comments. In addition, we thank the anonymous referees for their helpful comments.

**Conflicts of Interest:** The authors declare no conflict of interest. This is a scientific study using satellite imagery, which has no commercial goal.

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
