# Peer review of "Comparison of GF2 and SPOT6 Imagery on Canopy Cover Estimating in Northern Subtropics Forest in China"

_forests, doi:10.3390/f11040407_

Round 1
Reviewer 1 Report
The manuscript is clearly written and it is easy to understand. Furthermore, the motivation and objectives of the manuscript are good and worth publishing. However, several changes should be made to make the manuscript stronger.
1. The manuscript is about comparing SPOT-6 and Gaofen-2 satellite imagery in mapping forest canopy cover in central China. However, in the objectives, you write that the aim is to test the applicability of Gaofen-2 in mapping canopy cover. In discussion, you go back to discussing how Gaofen-2 is good for mapping canopy cover and its four bands are sufficient when compared to coarser spatial resolution imagery with more bands. In order to address this mismatch, you should redo the analysis with other satellite imagery, such as Landsat 8 and Sentinel-2 which are freely available. Therefore, you should compare four (or even more) different satellite images in mapping canopy cover. You could do separate regressions for the different satellite images and then do a regression in which you combine the different satellite images in the same regression (it has been shown multiple times that the inclusion of multiple data sources increases model performance). With a comparison like this, the empirical part would fit the introduction and discussion better. I am neither convinced that the comparison in Section 3.2 between GF-2 and SPOT-6 bands is necessarily needed in the results; the section 3.2 could maybe be removed. There is neither justification why you resampled SPOT-6, it seems to be an unnecessary step.
2. In the discussion, you also write of the benefits of texture features in boosting regression performance. However, you do not really test this as you do not do regressions separately for models with and without texture as a predictor variable. You do not either provide feature importance metrics which can be easily calculated with random forest. Furthermore, it is more than doubtful to write that standard deviation is a texture measure. Therefore, you should calculate texture with a more sophisticated method, e.g. GLCM, and then test whether texture is important by either calculating feature importance scores or by performing separate regressions.
3. One of the aims of the study is to compare ensemble regression methods to CART. There is no novelty in a comparison like this, and it has been shown multiple times that ensemble methods are better than simpler methods. Therefore, you should drop CART from the analysis. Furthermore, you do not specify what parameter values you used for the different regression methods. Generally, random forest is insensitive to changes in parameter values, but the situation is different for AdaBoost and GDBoost. You should definitely specify the parameter values and describe how you chose the optimal values. You should also specify what software was used for the statistical analysis.
4. One comparison was also testing whether inclusion of class variable (forest type) changes model performance. I would perhaps drop this analysis also. This complicates the manuscript a little and it does not help in addressing the main objectives of the manuscript. Furthermore, if the aim is to construct maps of canopy cover, the inclusion of forest type variable hinders the map construction (i.e. you should first construct a map of forest type). Similarly, it is a little confusing that you include a class variable in a regression in which other predictor variables are continuous. Instead, you could conduct regression separately for different forest types or for evergreen and deciduous trees. However, it seems that there are not enough field observations for such analysis.
5. You do not include any result maps in the manuscript. It would be good to include some maps of canopy cover and with the maps, you could also calculate landscape average for canopy cover.
6. You conducted field work in 97 plots, which is fine. It is also well described, in general, how canopy cover was estimated in this plots. However, the description lacks information about how the sampling was done. Was it a random sampling, stratified random sampling or something else? Additionally, what was the precision of the GPS? Did you verify that field work plots are located in the right positions in the different remote sensing images? How did you actually calculate the spectral properties for the field work plots. Did you calculate average values over the plots or did you pick value from just one pixel? Please specify.
7. There is some repetition in the manuscript. For instance, Tables 1 and 3 and Figure 3 repeat each other. I would keep only one of these illustrations. Moreover, the different spectral indices and other predictor variables should be listed in a table instead of text. Currently, there is a lot of repetition in Section 2.2.3.
8. After you have made the changes in methods and results, also the introduction and discussion could be amended a little. As already stated above, there are some mismatches between the empirical part and introduction/discussion. In particular, most of the discussion seem to be merely speculation, and many of the reported issues are not really tested in the manuscript (e.g. the value of infrared band and texture). Finally, I think that the manuscript would benefit from inclusion of more references.
Author Response
The manuscript is clearly written and it is easy to understand. Furthermore, the motivation and objectives of the manuscript are good and worth publishing. However, several changes should be made to make the manuscript stronger.
- The manuscript is about comparing SPOT-6 and Gaofen-2 satellite imagery in mapping forest canopy cover in central China. However, in the objectives, you write that the aim is to test the applicability of Gaofen-2 in mapping canopy cover. In discussion, you go back to discussing how Gaofen-2 is good for mapping canopy cover and its four bands are sufficient when compared to coarser spatial resolution imagery with more bands. In order to address this mismatch, you should redo the analysis with other satellite imagery, such as Landsat 8 and Sentinel-2 which are freely available. Therefore, you should compare four (or even more) different satellite images in mapping canopy cover. You could do separate regressions for the different satellite images and then do a regression in which you combine the different satellite images in the same regression (it has been shown multiple times that the inclusion of multiple data sources increases model performance). With a comparison like this, the empirical part would fit the introduction and discussion better. I am neither convinced that the comparison in Section 3.2 between GF-2 and SPOT-6 bands is necessarily needed in the results; the section 3.2 could maybe be removed. There is neither justification why you resampled SPOT-6, it seems to be an unnecessary step.
Response: Thanks very much for your comments. This paper focused on comparing SPOT-6 and Gaofen-2 satellite imagery in mapping canopy cover. So, Sentinel-2 and Landsat imagery were not considered in this paper, maybe next paper, we will focus on the multi sensor imageries in estimating CC. In order to put emphasis on paper object, we modified the introduction and discussion section. Meanwhile, according to the suggestion, we have removed the section 3.2, I agreed with the comments that we need not to compare the reflectance of SPOT6 and GF2. But I think the resampled SPOT6 to the same resolution of GF2 was necessary. Because the spectral variables and textural variables derived from imagery with the window size, if the resolution was not the same, it was difficult to compare it. Considering the resample errors, we only the neighborhood method to reduce its effects.
- In the discussion, you also write of the benefits of texture features in boosting regression performance. However, you do not really test this as you do not do regressions separately for models with and without texture as a predictor variable. You do not either provide feature importance metrics which can be easily calculated with random forest. Furthermore, it is more than doubtful to write that standard deviation is a texture measure. Therefore, you should calculate texture with a more sophisticated method, e.g. GLCM, and then test whether texture is important by either calculating feature importance scores or by performing separate regressions.
Response: Thanks very much for your comments. We agreed them. We have added the gray co-occurrence matrix to get eight textural variables. And, we also have compared the importance value from in estimating CC. The results have been added in the section 3.1. The discussion part has been revised according the results.
- One of the aims of the study is to compare ensemble regression methods to CART. There is no novelty in a comparison like this, and it has been shown multiple times that ensemble methods are better than simpler methods. Therefore, you should drop CART from the analysis. Furthermore, you do not specify what parameter values you used for the different regression methods. Generally, random forest is insensitive to changes in parameter values, but the situation is different for AdaBoost and GDBoost. You should definitely specify the parameter values and describe how you chose the optimal values. You should also specify what software was used for the statistical analysis.
Response: Thanks very much for your comments. We agreed them. We have dropped the CART model in estimating CC and put emphasizes on analysis the three ensemble models. The python package scikit-learn was used to implement the three models. The grid search method was used in optimize the best parameters in RF, AdaBoost and GDBoost modes. We have added the introduction in section 2.3 and showed the best parameters in section 3.2.
- One comparison was also testing whether inclusion of class variable (forest type) changes model performance. I would perhaps drop this analysis also. This complicates the manuscript a little and it does not help in addressing the main objectives of the manuscript. Furthermore, if the aim is to construct maps of canopy cover, the inclusion of forest type variable hinders the map construction (i.e. you should first construct a map of forest type). Similarly, it is a little confusing that you include a class variable in a regression in which other predictor variables are continuous. Instead, you could conduct regression separately for different forest types or for evergreen and deciduous trees. However, it seems that there are not enough field observations for such analysis.
Response: Thanks very much for your comments. We agreed them. We have removed the analysis the forest type on model performance. It was difficulty to accurate map forest type in large complex area. This paper was focused on mapping CC, adding forest type maybe improve results, but it was not the main object.
- You do not include any result maps in the manuscript. It would be good to include some maps of canopy cover and with the maps, you could also calculate landscape average for canopy cover.
Response: Thanks very much for your comments. We agreed them. We have added the whole map of CC with highest accuracy model in this study area, and added the statistic information about this area. The results addressed in Section 3.3.
- You conducted field work in 97 plots, which is fine. It is also well described, in general, how canopy cover was estimated in this plots. However, the description lacks information about how the sampling was done. Was it a random sampling, stratified random sampling or something else? Additionally, what was the precision of the GPS? Did you verify that field work plots are located in the right positions in the different remote sensing images? How did you actually calculate the spectral properties for the field work plots. Did you calculate average values over the plots or did you pick value from just one pixel? Please specify.
Response: Thanks very much for your comments. We agreed them. the stratified sampling method was used in this investigation. The position of plots was located by a differential GPS unit (Trimble GeoXH6000 GPS units) and corrected with high precision real-time differential signals received from Hubei Continuously Operation Reference Stations (HBCORS). The location error was less than 1m, which allowing for the plots to be good geo-referenced with satellite data. Geometrically correction was applied before atmospheric correction. The root mean square error of geometrically correction was 1.83m, which was lower than 1 pixel. According the spatial resolution and plot size, we derived each variable with the median value of a 5 × 5 window. These details have been added in section 2.
- There is some repetition in the manuscript. For instance, Tables 1 and 3 and Figure 3 repeat each other. I would keep only one of these illustrations. Moreover, the different spectral indices and other predictor variables should be listed in a table instead of text. Currently, there is a lot of repetition in Section 2.2.3.
Response: Thanks very much for your comments. We agreed them. we have deleted the Table 3 and Figure 3 in results part and add the canopy cover characteristic information the section 2.2.3. Because the reviewer 1 also pointed out this problem and suggested the repetition with the section 2.2.3. We combined these two suggestions.
- After you have made the changes in methods and results, also the introduction and discussion could be amended a little. As already stated above, there are some mismatches between the empirical part and introduction/discussion. In particular, most of the discussion seem to be merely speculation, and many of the reported issues are not really tested in the manuscript (e.g. the value of infrared band and texture). Finally, I think that the manuscript would benefit from inclusion of more references.
Response: Thanks very much for your comments. We agreed them. In order to put emphasis on paper object, we modified the introduction and discussion section.
Thanks very much for your good comments.
Reviewer 2 Report
The results presented in the manuscript “Comparison of GF2 and SPOT6 imagery on Canopy Cover Estimating in Northern Subtropics Forest in China” by Jingjing Zhou, Yuanyong Dian, Xiong Wang, Chonghuai Yao, Yongfeng Jian, Yuan Li, Zeming Han are valuable and maybe of international interest. The authors demonstrated a good knowledge of the problematic and of the related literature. The manuscript is short and concise.
However, there are several smaller issues:
Line 7 - 11: Check again punctuation. Edit hyperlinks!
Line 79 - 81: In accordance with scientific standards and International Code of Botanical Nomenclature (ICBN), scientific plant names, irrespective of rank, should be given in italics. The author’s name should be written in normal print at least once, when mentioned for the first time in the text or in a table, and should be omitted subsequently. They should be abbreviated in conformity with the Authors of Plant Names, easily accessible on https://www.ipni.org/ or http://www.theplantlist.org/. After the first mention, the generic name should be abbreviated to its initial.
Line 199: Table 3 is unnecessary. Describe the results in the text.
Please, check again punctuation.
Use a comma after the penultimate item in a list of three or more items, before 'and' or 'or' (Oxford comma).
Author Response
The results presented in the manuscript “Comparison of GF2 and SPOT6 imagery on Canopy Cover Estimating in Northern Subtropics Forest in China” by Jingjing Zhou, Yuanyong Dian, Xiong Wang, Chonghuai Yao, Yongfeng Jian, Yuan Li, Zeming Han are valuable and maybe of international interest. The authors demonstrated a good knowledge of the problematic and of the related literature. The manuscript is short and concise.
However, there are several smaller issues:
Line 7 - 11: Check again punctuation. Edit hyperlinks!
Response: we have made correction of the hyperlinks on emails.
Line 79 - 81: In accordance with scientific standards and International Code of Botanical Nomenclature (ICBN), scientific plant names, irrespective of rank, should be given in italics. The author’s name should be written in normal print at least once, when mentioned for the first time in the text or in a table, and should be omitted subsequently. They should be abbreviated in conformity with the Authors of Plant Names, easily accessible on https://www.ipni.org/ or http://www.theplantlist.org/. After the first mention, the generic name should be abbreviated to its initial.
Response: we have checked the scientific plant names in method part and modified the name according the website reviewer provide.
Line 199: Table 3 is unnecessary. Describe the results in the text.
Response: We have deleted the Table 3 and Figure 3 in results part and add the canopy cover characteristic information the section 2.2.3. Because the reviewer 2 also pointed out this problem and suggested the repetition with the section 2.2.3. We combined these two suggestions.
Please, check again punctuation.
Use a comma after the penultimate item in a list of three or more items, before 'and' or 'or' (Oxford comma).
Response: We have revised the whole manuscript carefully and tried to avoid any grammar or syntax error.
Thanks very much for your good comments.
Round 2
Reviewer 1 Report
The manuscript has improved significantly from the previous version. There are only some relatively minor issues that should be improved.
First, I feel that a proper proof-reading is needed, in particular new additions need language checking.
Second, the stratified sampling still needs some extra information. Did you use a map which defined the strata or how the size of the strata was set?
Third, I would put the field data characteristics (l138-143 and Table 1) into results. However, you should still describe how ANOVA was carried out in methods but add its results into results section.
Fourth, methodological description of parameter optimization (l478-486) should be in methods. Its results can still be in results. Did you use the same optimal parameter values for each ensemble methods (i.e. same number of trees for all three methods and same learning rate for AdaBoost and GDBoost)? Did you not tune parameter values separately for different ensembles and for different satellite images?
Author Response
The manuscript has improved significantly from the previous version. There are only some relatively minor issues that should be improved.
First, I feel that a proper proof-reading is needed, in particular new additions need language checking.
Response: Thanks very much for your comments. We have revised the whole manuscript carefully and tried to avoid any grammar or syntax error
Second, the stratified sampling still needs some extra information. Did you use a map which defined the strata or how the size of the strata was set?
Response: Thanks very much for your comments. The strata ratio was determined ty the area ratios of different forest type area to total forested area. The area ratios of needle forest, broadleaf forest and mixed forest to all forested area were 42.3%, 24.7% and 33.0% respectively according to statistical data from local forest management department. I have added this information in section 2.1 and section 2.2.
Third, I would put the field data characteristics (l138-143 and Table 1) into results. However, you should still describe how ANOVA was carried out in methods but add its results into results section.
Response: Thanks very much for your comments. We agreed them. We have moved the field data characteristics into 3.1.
Fourth, methodological description of parameter optimization (l478-486) should be in methods. Its results can still be in results. Did you use the same optimal parameter values for each ensemble methods (i.e. same number of trees for all three methods and same learning rate for AdaBoost and GDBoost)? Did you not tune parameter values separately for different ensembles and for different satellite images?
Response: Thanks very much for your comments. The grid search was done separately for each image and model. In AdaBoost and GDBoost, we tuned two parameters: ntree and learning_rate, while we tuned ntree and max_features in RF model. The ntree was tested from 50 to 500 stepped by 50, the learning_rate was tested 0.5 to 1.0 stepped by 0.1, and the max_features was tested from 2 to 17 stepped by 1. R2 heat map of search space with different models and images were shown in Fig.4. We have added the results in section 3.2.